# Span-based single-stage joint entity-relation extraction model

**Dongchen Han**◯, **Zhaoqian Zheng**◯, **Hui Zhao**◯*, **Shanshan Feng, Haiting Pang**

School of Computer Science and Engineering, Changchun University of Technology, Changchun, Jilin, China

◯ These authors contributed equally to this work.
* 2202003049@stu.ccut.edu.cn

## Abstract

Extracting entities and relations from the unstructured text has attracted increasing attention in recent years. The existing work has achieved considerable results, yet it is difficult to solve entity overlap and exposure bias. To address cascading errors, exposure bias, and entity overlap in existing entity relation extraction approaches, we propose a joint entity relation extraction model (SMHS) based on a span-level multi-head selection mechanism, transforming entity relation extraction into a span-level multi-head selection problem. Our model uses span-tagger and span-embedding to construct span semantic vectors, utilizes LSTM and multi-head self-attention mechanism for span feature extraction, multi-head selection mechanism for span-level relation decoding, and introduces span classification task for multi-task learning to decode out the relation triad in a single-stage. Experiments on the classic English dataset NYT and the publicly available Chinese relationship extraction dataset DuIE 2.0 show that this method achieves better results than the baseline method, which verifies the effectiveness of this method. Source code and data are published here (https://github.com/Beno-waxgourd/NLP.git).

**Data Availability Statement:** All relevant data are within the paper and its Supporting information files.

**Funding:** This study was supported by the "13th Five-Year Plan" Science and Technology Project

## Introduction

Relation Extraction (RE) [1], as one of the important subtasks of information extraction, aims to identify entities and relations between entities from unstructured text and finally form a triad such as (subject, relation, object). The traditional pipeline approach divides entity relationship extraction into two subtasks. Named Entity Recognition (NER) and Relationship Classification (RC). However, since these two subtasks are independent of each other, the intrinsic connections and dependencies between the two subtasks can be overlooked and are prone to entity overlap and exposure bias. The exposure bias problem arises due to the inconsistent distribution of the real labels used in model training and the model-generated labels used in prediction. Entity overlap refers to some identical entities between different relational triples of a sentence, as shown in Table 1.

Unlike pipelined approaches, joint extraction can identify entities and entity relationships with a single model by exploiting the close interaction information between entities and

(JJKH20200677KJ) of the Department of
Education of Jilin Province, China.

**Competing interests:** The authors have declared
that no competing interests exist.

**Table 1. Examples of entity overlap.**

| Entity Overlap Type | Text | Triplets |
|---|---|---|
| Normal | Obama was born in the United States. | (Obama, birthplace, the United States) |
| SEO(Single Entity Overlap) | Obama was born in the United States and graduated from Harvard University. | (Obama, birthplace, the United States) (Obama, graduated from, Harvard University) |
| EPO(Entity Pair Overlap) | Washington D.C. is the capital of the United States. | (the United States, contains, Washington D.C.) (the United States, capital city, Washington D.C.) |

relationships. However, it is still difficult to solve the problems of entity overlap and exposure bias simultaneously.

This paper proposes a joint relation extraction model(SMHS) based on a span multi-head selection mechanism to address the difficulty of simultaneously solving entity overlap and exposure bias concerning extraction. This model treats relation extraction as a span-level multi-head relation selection problem, and the model is divided into two parts as a whole: encoding and decoding. The encoding layer uses span-tagger and span-embedding to construct span-level semantic vectors and uses LSTM and multi-head self-attention for deep span feature extraction; the decoding layer uses a span classifier and multi-head selection mechanism for span classification and relational decoding. The span vectors constructed by the model are independent of each other, which naturally solves the problem of nested entities; combined with the span classification task, the span type information is indirectly incorporated by sharing the span vectors, which enhances the type constraint of the span relation; the relation multi-head selection between the span vectors can directly decode the spans and the relation between the spans and realize the single-step decoding of the relation triad, thus simultaneously solving the problems of entity overlap and exposure bias.

The model was tested on the classic English dataset NYT and the authoritative Chinese public dataset DuIE 2.0 respectively. The results show that compared with the baseline model, the model has significant improvement, and performs well in dealing with SEO and EPO, which proves that the model is effective in solving the problems of entity overlap, error accumulation, exposure deviation, etc.

## Related work

Currently, the mainstream modeling approaches for joint extraction are divided into four main types: (a) Modeling entity relation extraction as end-to-end table-filling. Gupta et al. [2] proposed a relation extraction method based on table-filling. However, this method allows entities and entity-relations to share parameters in a model, extracts entities and relations separately, generating redundant information, and has high computational complexity. (b) Modeling entity-relation extraction as sequence-labeling. Zheng et al. [3] first unified entity identification and relationship identification into a single task of sequence-labeling. This approach achieves single-step extraction but cannot overcome the entity overlap problem since it can only assign one label to each token when performing annotation. Dai et al. [4] used a multi-round labeling scheme on this basis to solve the entity overlap problem. Such labeling-based approach requires manual design of an ingenious annotation system. (c) Modeling entity relation extraction as an Encoder-Decoder structure [5, 6], which can cope with the entity overlap problem with good computational complexity. However, it is more difficult to solve the exposure bias and nested entity problems. (d) Modeling entity-relation extraction as a relation mapping between subject entity and object entity. Wei et al. [7] proposed a cascaded extraction framework first to extract principal candidate entities that may involve target

relations and then label the corresponding object relations and guest entities for each extracted principal entity. This approach solves the entity overlap problem while failing to address the exposure bias problem.

In addition, Miwa et al. [8] attempted to use a joint extraction method based on shared parameters and used syntactic dependency trees and LSTM to encode entities. However, it still causes entity redundancy and cannot solve the entity overlap problem. Bekoulis et al. [9] viewed relation extraction as a multi-headed selection problem and used tag embedding to incorporate tag features, but decoded token-level relations, which require joint entity boundaries identified by the recognition module to decode entity relations, which will have error accumulation jointly. Katiyar et al. [10] used a pointer network to decode entity relations. However, there is an omission of entities, and the entity boundary information is not fully utilized, which can lead to a distraction of attention mechanisms. LI et al. [11] transformed the relation extraction into a multi-round dialogue task, which can capture hierarchical dependencies well, and the design problem can encode important prior relation information, which is helpful for entity relation extraction. A reinforcement learning mechanism is used to alleviate the error accumulation problem. Dixit et al. [12] use span alignment and span filterer to generate candidate spans, but the final span relation classification generates redundancy, which leads to accuracy degradation. Fu et al. [13]proposed an end-to-end joint extraction model based on graph convolutional neural network (GCN), which takes into account the interaction between named entities and relations to extract relations through relation-weighted GCN better. Wang et al. [14] proposed a handshake-tagging and token-linking-based approach, which decodes entity-relation triples in a single step but still requires combining the recognition results of multiple tokens for relation decoding. Sui et al. [15] treated joint entity and relation extraction as an ensemble prediction problem, used a non-autoregressive decoder for the previous problem of decoding triples in a given order, and used a bipartite graph matching loss function, but was limited by a predetermined number of triples hyperparameters. Zheng et al. [16] designed an extraction method using dynamic sliding window, and designed three mapping strategies to combine and arrange tokens and then tile them into a span sequence. However, the extracted entities with the same relationship label adopt the nearest matching method, which reduces the accuracy.

In general, most mainstream relation extraction models can hardly solve entity overlap and exposure bias simultaneously. In contrast, a small number of models can solve the above two problems but have corresponding drawbacks, such as not using entity type information and less interaction between entities. At present, there is still more room to explore the problems of entity overlap and exposure bias with extraction models.

## Model

This section divides the model into three parts: the span-tagger, the encoding, and the decoding, for a detailed description. The overall framework of the model is shown in Fig 1.

### Span-tagger

This model has designed a method called a span-tagger, which combines the position indexes of the first and last word elements of the span into an index tuple to tag the span. By enumerating the span, all enumerated spans cover all candidate entities, so that the problem of entity nesting can be solved. The essence of span-tagger is to tag the mapping relationship between the tagged span and its position in the span embedding matrix, that is, to map the position of the original span word to obtain the position in the span embedding matrix. For example: "Obama was born in the United States". Where "Obama" is tagged as (0, 0) and the span

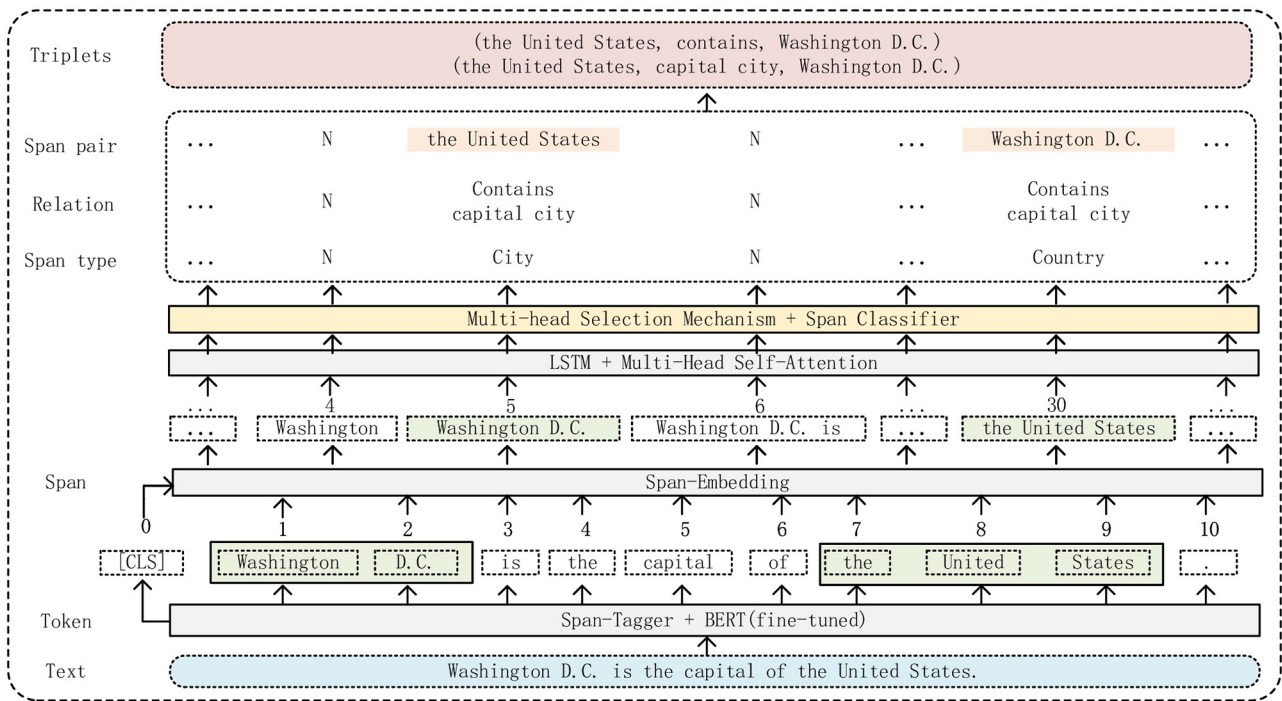

**Fig 1. The framework of SMHS.**

position index is 0, "Obama was" is tagged as (0, 1) and the span position index is 1.., and so on. The tagging sequence of span uses three different strategies for location mapping, namely, the same start mapping strategy, the same end mapping strategy and the same length mapping strategy. See Fig 2 for details.

Suppose the length of the input sequence is **n** and since the sentences are divided by individual characters, the number of tokens is also **n**. Given the temporal and semantic nature of the text, each token can only form a span from the tag in the position before it. With the length of the text sequence, if all spans are enumerated, the number of spans increases rapidly. It contains a large number of useless spans, reducing accuracy and consuming much computational power. So a window length is set **w** to filter the useless spans that exceed the window length, then the number of all spans in the sequence of token **m** are:

$$m = \frac{(2n+1)w - w^2}{2}, 0 < w \leq n \tag{1}$$

Construct a mapping **D** that transforms the position index tuples of the head and tail token of a span to the indexes of the corresponding span embedding matrix. Assume that the position index of the head token of a span is **i** and the position index of the tail token is **j**. The resulting token position index tuple is **(i, j)**, which is in the **k**th row of the span embedding lookup table, then we have:

$$D((i,j)) = k, 0 < i \leq j \leq n, 0 \leq k < m \tag{2}$$

The span is an arrangement of several consecutive tokens. The spans within the window length of each character are enumerated, so the candidate entities and relations must also be in the enumerated spans, which can solve the problem of entity nesting and error accumulation

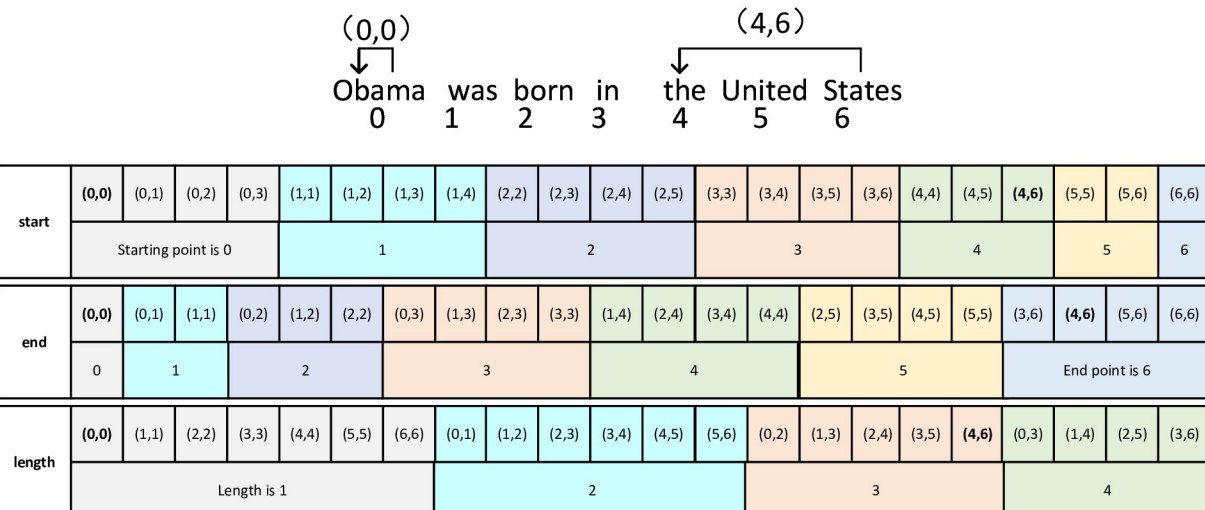

**Fig 2. Example of span-tagger.** (window length is 4). The value in the matrix is the tag value of span, and its position is the index of span. According to different span mapping strategies, the specific location index in the span embedding matrix is different. Span mapping strategy with the same starting point: fix the starting point of the span in the text, change the span length by changing the ending point of the span, until the span length reaches the maximum length of the window, and train the spans with the same starting point together, as shown in the first line of the figure; The mapping strategy for spans with the same endpoint is just the opposite: tag the span by fixing the endpoint of the span, changing the starting point of the span, and train the spans with the same starting point together, as shown in the second line of the figure; Mapping strategy of the same span length: the fixed window length traverses the text, and then increases the window length to traverse the text again until the maximum window length is reached, and the spans with the same length are trained together, as shown in the third line of the figure.

more perfectly. The span tagger enables the token position index tuple to correspond to its constituent span position indexes quickly. It enables the subsequent construction of the span semantic vectors to be combined into a span-embedding matrix according to the results of the span-tagger, which facilitates the subsequent selection of span multi-headed relations.

## Encoding

**Token representation.** The character-level BERT (Bidirectional Encoder Representation from Transformers) [17–19] language model is used to encode the utterance with tokens, extracting contextual semantic information for each character and the overall semantic information of the sentence. Assuming that the sentence input is represented as $\mathbf{S} = [s_1, s_2, s_3, \ldots, s_n]$, then the vector after BERT encoding is represented as:

$$H, \bar{H} = BERT(S), H \in \mathbb{R}^{n*d}, \bar{H} \in \mathbb{R}^d \tag{3}$$

Where **H** is denoted as the vector of token generated after each token is encoded by BERT, the **H̄** is the vector of sentences generated after the whole sentence is encoded by BERT.**n** is the length of the sequence, and **d** is the dimension of the BERT hidden state.

**Span representation.** The construction of span semantic vector and span embedding matrix is the core of the model. Previously, span based models were used to extract span features indirectly. However, this model constructs span semantic vector by averaging pooling each word element vector that makes up the span and splicing it with the sentence vector, which can directly extract span features.

Suppose $\mathbf{H}_{i:j} = [h_i, h_{i+1}, \ldots, h_j]$, $\mathbf{H}_{i:j} \in \mathbb{R}^{(j-i+1)*d}$, means that the window length is **w** with position index **i** of the head token to the middle of the tail token with position index of **j** The

semantic vector of the span composed is calculated of it is as follows:

$$k = D((i,j)), 0 < i \leq j \leq n \tag{4}$$

$$x'_k = Concat(Meanpool(H_{i:j}), \bar{H}),$$
$$x'_k \in \mathbb{R}^{2d} \tag{5}$$

$$x_k = Tanh(W_h x'_k + b_h), x_k \in \mathbb{R}^d \tag{6}$$

where $W_h \in \mathbb{R}^{d*2d}$ is a parameter matrix and $b_h \in \mathbb{R}^d$ is a bias vector to be learned during training.**k** is the position in the span-embedding matrix of the span composed by the head token position index **i** to the tail position index **j**, and $\mathbf{x}_k$ denotes the span semantic vector of the composed spans.

The process of Eqs 4 to 6 is carried out cyclically to construct all span semantic vectors, and then the span embedding matrix is composed in order according to the mapping result of the span-tagger. $\mathbf{X} = [x_1, x_2, x_3, \ldots, x_m], X \in \mathbb{R}^{m*d}$, denoted as the span embedding matrix composed of all span semantic vectors.

**Span feature extraction.** LSTM and multi-headed self-attention mechanism are used to enhance span information interaction and deep span feature extraction.

LSTM [20] controls the input content and the content stored in the memory cell through input gates, forgetting gates, and output gates to form a memory of the previous input information, which can effectively solve the gradient explosion and gradient disappearance problems. The span embedding matrix is encoded by LSTM, which can effectively enhance the information interaction between spans and capture the dependency relation between spans.

Attention mechanism can selectively focus on important information of text. Multi-headed self- attention mechanism [21] is a variation of attention mechanism, in which Q (query), K (key) and V (value) are equal. Multiple queries are used to extract multiple groups of different information from input information in parallel for splicing, and common attention is paid to information from different representation subspaces at different locations, which helps the model capture more span feature information.

Assume $\mathbf{P} = [p_1, p_2, p_3, \ldots, p_m]$, denoted as the spans vector after LSTM encoding, then we have:

$$P = LSTM(X), P \in \mathbb{R}^{m*d} \tag{7}$$

Assume that $\mathbf{T} = [t_1, t_2, t_3, \ldots, t_m]$, denoted as a span vector after encoding by the multi-headed attention mechanism and using $z$ parallel heads are used to capture the features, then the $i$th head is computed as follows:

$$Head_i = softmax\left(\frac{(QW_i^Q)(KW_i^K)}{\sqrt{d/z}}\right)(VW_i^V) \tag{8}$$

$$T = Concat(head_1, head_2, \ldots, head_z)W^o,$$
$$T \in \mathbb{R}^{m*d} \tag{9}$$

where $W_i^Q \in \mathbb{R}^{d*d/z}$, $W_i^K \in \mathbb{R}^{d*d/z}$, $W_i^V \in \mathbb{R}^{d*d/z}$, $W^o \in \mathbb{R}^{d*d}$, the $W^o \in \mathbb{R}^{d*d}$ is the trainbale parameter matrix.

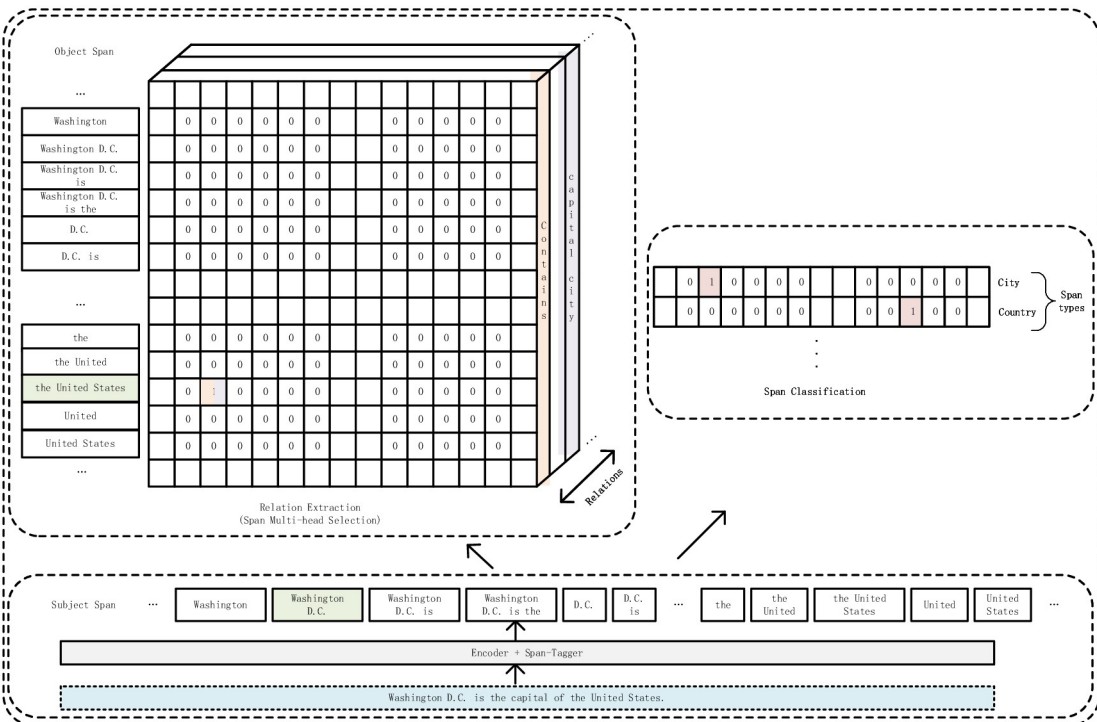

**Fig 3. Example of decoding process (window length is 4).** On the left is the decoding of relation extraction, and on the right is the decoding of span classification. The left three-dimensional matrix row is the index of the subject, the column is the index of the object, and the side surface is the index of the relationship.A cell with a value of 1 in the matrix represents a triple of subject, object and relationship corresponding to its coordinates. For example, the triples in the figure are (the United States, contains, Washington D.C.) and (the United States, capital city, Washington D.C.).

## Decoding

The decoding is responsible for predicting two subtasks. Namely, relation extraction (RE) and span classification (SC), as shown in the Fig 3, which will be presented below. Fig 4 shows the flattened matrix on the left side of Fig 3.

**Relation extraction.**   The model uses the span-level multi-headed selection mechanism [9] for relationship extraction, which directly performs multi-headed relationship selection between spans without entity recognition, and extracts relationship triples in a single step.

Assume $\mathbf{R} = \{r_1, r_2, \ldots, r_l\}$ that is denoted as the set of relations and $\mathbf{l}$ is the number of relations. $\tilde{S}(t_i, t_j, r_k)$ denote the score of the $k$-th relation between span pair $(\mathbf{t}_i, \mathbf{t}_j)$. The calculation formula is as follows:

$$\tilde{S}(t_i, t_j, r_k) = V_k tanh(U_K t_i + W_k t_j + b_k) \tag{10}$$

where $V_k \in \mathbb{R}^d, U_k \in \mathbb{R}^{d*d}, W_k \in \mathbb{R}^{d*d}, b_k \in \mathbb{R}^d$, are parameters for the $k$-th relation. Next, the probability of span $t_i$ selected as the head of $t_j$ with the relation $r_k$ is calculated as:

$$P_r(head = t_i, relation = r_k | t_j) = \sigma(\tilde{S}(t_i, t_j, r_k)) \tag{11}$$

**Span classification.**   Entity information, especially category information, has been shown to help improve the effectiveness of relation extraction models [22], and there are entity category constraints in the relation recognition process. For example, the relation "singer" can

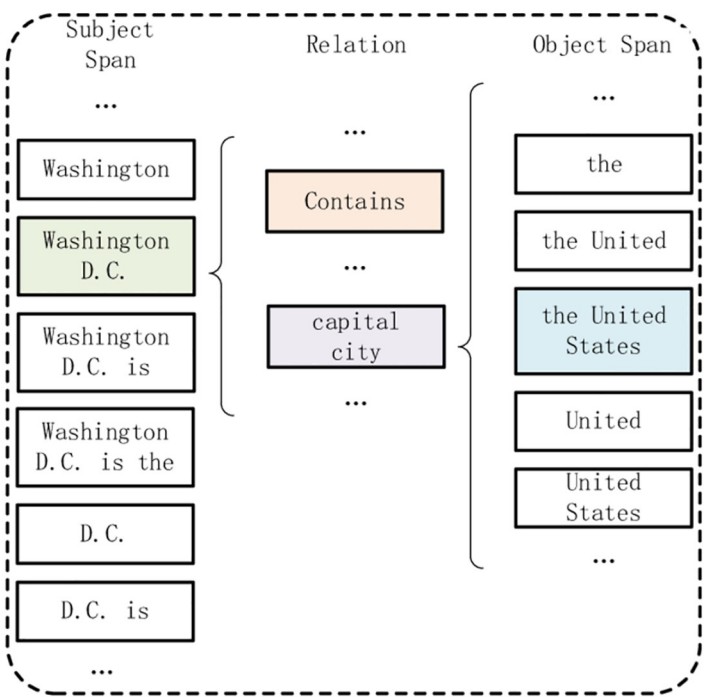

**Fig 4. Flattened matrix of relation extraction of Fig 3.**

only be composed of entities with entity types "figure" and "song". In order to utilize entity type and entity type constraint information, this method indirectly introduces entity type and entity type constraint information through shared span coding, which is used for multi-task learning of span type classification task and auxiliary training of span relation extraction task. Multi-task learning [23] can improve the generalization and robustness of the model by using the interaction between multiple tasks and the special information contained through shared encoding.

Assume $\mathbf{I} = \{i_1, i_2, \ldots, i_g\}$ that is the set of span types, and $\mathbf{g}$ is the number of span types. $\bar{S}(t_j, i_k)$ is denoted span $t_j$ is the score of the $k$-th type. The calculation formula is as follows:

$$\bar{S}(t_j, i_k) = Z_k t_j + b'_k \tag{12}$$

where $Z_k \in \mathbb{R}^d, b'_k \in \mathbb{R}^1$, are parameters for the $k$-th type. Next, the probability of span $t_j$ is the $k$-th type is calculated as:

$$P_t(type = i_k | t_j) = \sigma(\bar{S}(t_j, i_k)) \tag{13}$$

## Loss function

The loss function for relation extraction is defined as follows:

$$Loss_{re} = \sum_{i=1}^{m}\sum_{j=1}^{m}\sum_{k=1}^{l} - logP_r(head = t_i, relation = r_k | t_j) \tag{14}$$

The loss function for span classification is defined as follows:

$$Loss_{sc} = \sum_{j=1}^{m}\sum_{k=1}^{g} -logP_t(type = i_k|t_j) \tag{15}$$

The final loss function is defined as follows:

$$Loss_{final} = \alpha * Loss_{re} + \beta * Loss_{sc} \tag{16}$$

where $\alpha$, $\beta$ is the loss function weight, which is used to balance the two tasks. Our model is implemented with PyTorch and the network weights are optimized with the *Adam* [24] optimizer.

## Experiments

### Dataset

For the convenience to compare our model with previous work, we follow the popular choice of datasets:

(a) NYT: A dataset widely used in relation extraction tasks. This dataset is generated by aligning the relationships in freebase with the New York Times (NYT) corpus through remote supervision. It contains more than 100000 triples, 50000 English texts, 24 relationship types, and 1 entity type.

(b) DuIE2.0(https://aistudio.baidu.com/aistudio/competition/detail/46/0/datasets): The largest schema based Chinese relationship extraction data set in the industry, which belongs to Baidu company. It contains more than 430000 triple data, 210000 Chinese sentences, 48 predefined relationship types, 43 simple relationship types and 5 complex relationship types. The sentences in the dataset come from the texts of Baidu Encyclopedia, Baidu tieba and Baidu feed flow. The data annotation is generated through manual annotation and remote supervision.

Since the relationship between enumeration span and enumeration span will occupy a large amount of storage space, texts larger than 62 and entities smaller than 15 will be deleted from DuIE2.0 training set or test set respectively to form new training and test sets. The NYT dataset is not modified. The distribution of the new training set and test set is the same as that of the original dataset. To ensure the fairness of the data, all the comparison tests in this article will be conducted in the new dataset. The statistical information of the dataset is shown in Table 2.

### Setting

The GPU used in this experiment is NVIDIA TITAN XP * 4, the operating system is CentOS 7.9, the programming language is Python 3.7, and the deep learning framework is Pytorch 1.8. The pre-training models are bert-base-cased and 768-dimensional Chinese character-level

**Table 2. Dataset information.**

| Dataset | | Normal | SEO | EPO | Texts | Triples | Max length of text | Max length of entity |
|---|---|---|---|---|---|---|---|---|
| NYT | Train | 41109 | 8918 | 15606 | 56195 | 112936 | 378 | 9 |
| | Test | 3266 | 1297 | 978 | 5000 | 10142 | 137 | 7 |
| DuIE | Train | 59464 | 26999 | 3338 | 89801 | 144858 | 61 | 15 |
| | Test | 7208 | 3257 | 395 | 10860 | 17565 | 76 | 15 |

BERT. In the English model, because the text length is too long, if we set a larger window length, it will cause an unbearable cost of time and space, so our window length is set to 6, and the maximum sequence length is 128. In the Chinese model, the sliding window length is 17, and the maximum sequence length is 64. The number of attention heads is the same as the pre-training model, the mapping strategy uses the mapping method with the same starting point, the learning rate is 0.00002, the epoch is 20, $\alpha$: $\beta$ = 3: 1.

## Evaluation

In this paper, the accuracy, recall and F1 values are used to evaluate the experimental results, and strict criteria are adopted: when the primary entity, primary entity type, customer entity, customer entity type and relationship type are judged correctly, the extraction result is considered correct.

## Experimental results and analysis

In order to verify the effectiveness of the model, two groups of experiments are designed in this paper: validation experiment and hyperparameter comparison experiment. Five baselines were selected for the validation experiment; The hyperparameter contrast experiment is designed according to different mapping strategies, window length and other hyperparameter of the model.

**Validation experiments.** The comparative baseline models used in this paper are: (a) Novel-Tagging [3], a model that transforms the two tasks of relation extraction and entity extraction into a unified sequence annotation. (b) GraphRel [13], an end-to-end joint extraction model based on GCNs. (c) CMAN [25], a deep cross-modal attention network model for label information and entity feature information. (d) TP-Linker [14], a single-stage joint extraction model by linking token and token annotations. (e) SLM [16], an entity-relationship extraction model based on Span labeling.

All the above models use the same BERT pre-training model for token encoding of sentences. The experimental results are shown in Table 3.

Table 3 shows that the SMHS method proposed in this paper has achieved the most advanced level in precision, recall and F1 scores. The first four baseline models are all decoded based on the word element vector, which has the disadvantage of unclear entity boundary recognition. Among them, novel tagging cannot overcome the problems of overlapping entities and nested entities, resulting in a low recall rate; GraphRel ignores the correlation between the two tasks, and the prediction of the relationship between entity pairs composed of multiple words may conflict, so the effect is not good; Although CMAN model uses the cross modal method to combine the label information, it does not solve the problems of overlapping entities and exposure bias in essence, resulting in its F1 on DuIE dataset being 5.2% lower than that of our model; Compared with the same single-stage decoding model TP-linker on NYT, our model failed to cover all entities during training because we could not set too long window length, so the effect was slightly inferior, but on the DuIE dataset, we achieved full coverage of entities, the precision rate increased by 2.5%, the recall rate increased by 3.3%, and F1 increased by 3.0%. This is because TP-linker classifies according to the semantic vector combined by the head and tail lexical elements, while other lexical elements between spans do not participate, resulting in insufficient semantic information fusion and the use of type information. Although SLM uses the span labeling method, the extracted entities with the same relationship labels use the nearest matching method, which leads to the F1 of the model on the two datasets is 8.9% and 11.4% lower than that of our model respectively, especially the value of SEO is 13.0% and 13.6% lower.

**Table 3. Comparison of experimental results.**

| Model | NYT | | | | | | DuIE | | | | | |
|---|---|---|---|---|---|---|---|---|---|---|---|---|
| | Normal | SEO | EPO | Prec. | Rec. | F1 | Normal | SEO | EPO | Prec. | Rec. | F1 |
| Novel-Tagging | - | - | - | 32.8 | 30.6 | 31.7 | 54.0 | 42.1 | 31.5 | 75.8 | 36.7 | 49.4 |
| GraphRel | 69.6 | 51.2 | 58.2 | 63.9 | 60.0 | 61.9 | - | - | - | 52.2 | 25.8 | 34.5 |
| CMAN | - | - | - | - | - | - | 74.5 | 69.3 | 66.1 | 77.3 | 68.1 | 72.4 |
| TP-Linker | **90.1** | **93.4** | **94.0** | 91.4 | **92.6** | **92.0** | 74.8 | 73.1 | 73.0 | 78.4 | 70.5 | 74.2 |
| SLM | 81.3 | 72.6 | 75.4 | 86.1 | 75.1 | 80.2 | 68.1 | 63.4 | 68.2 | 71.3 | 61.1 | 65.8 |
| SMHS | 89.2 | 87.3 | 88.4 | **97.5** | 82.0 | 89.1 | **77.3** | **77.0** | **76.6** | **80.9** | **73.8** | **77.2** |

The default hyperparameter values are: mapping = start, window_length = 6(NYT) and 15(DuIE), $\alpha$:$\beta$=3:1

In contrast, the model in this paper incorporates all token vectors and sentence vectors that constitute the spans, fully exploits token and sentence information, and performs well in SEO and EPO data. The experimental results show that the proposed model can effectively solve the entity overlap and exposure bias problems.

**Hyperparameter comparison experiment.** This section mainly studies the influence of different span index mapping strategies, different maximum window lengths and different loss function weights on the experimental results. All experiments in this section are conducted under the premise of a fixed learning rate.

*(a) Mapping strategies.* It is easy to see from Table 4 that when span tagging is performed, mapping strategies with the same starting point can obtain better results.

*(b) Window length.* The maximum length of entities in the dataset can be queried from Table 2. As shown in the Table 5, when the window length is less than the entity length, span tag cannot completely extract all entities, so the results will also be affected. At the same time, it can be seen that when the window length is greater than the entity length, the model can cover all entities in the sample. If the window length continues to increase, the score of the model will increase.

*(c) $\alpha$ and $\beta$.* Table 6 shows that if the weight of relation extraction in the loss function is appropriately increased and the weight of span classification is appropriately reduced, the model can get higher scores.

**Ablation experiments.** In order to further explore the influence of design methods, ablation experiments were conducted. The experiments are to remove Span-Classify; Remove the Multi-head Selection; Use Shared-Token representation instead of span representation. The results of the experiments are shown in Table 7.

The results show that the remove the Span-Classify results in the decrease of F1 value, indicating that span type information introduced by span type classification is effective for relation

**Table 4. Comparison experiment results of mapping strategy.**

| Dataset | Mapping Strategies | Prec. | Rec. | F1 |
|---|---|---|---|---|
| NYT | Start | 95.7 | 80.5 | 87.5 |
| | End | 94.7 | 81.2 | 87.4 |
| | Length | 89.1 | 74.9 | 81.4 |
| DuIE | Start | 78.9 | 73.0 | 75.8 |
| | End | 80.3 | 69.3 | 74.4 |
| | Length | 80.4 | 59.0 | 68.1 |

Other hyperparameter values: window_length = 6(NYT) and 15(DuIE), $\alpha$:$\beta$=1:1.

**Table 5. Comparison experiment results of mapping strategy.**

| Dataset | Window Length | Number of Entities | Coverage of Entities | Prec. | Rec. | F1 |
|---|---|---|---|---|---|---|
| NYT | 3 | 200814 | 0.9687 | 96.5 | 71.1 | 81.9 |
| | 6 | 207206 | 0.9995 | 95.7 | 80.5 | 87.5 |
| | 9 | 207292 | 1 | - | - | - |
| DuIE | 13 | 225130 | 0.9903 | 77.7 | 71.6 | 74.5 |
| | 15 | 226614 | 0.9968 | 78.9 | 73.0 | 75.8 |
| | 17 | 227332 | 1 | 78.9 | 73.9 | 76.3 |

Other hyperparameter values: mapping = start, $\alpha:\beta$=1:1.

extraction model, because for each relationship, only a specific span category can form this type, and span type classification task can introduce this type of information; The removal of Multi-head Selection leads to the decrease of F1 value of the model, which indicates that the design of three-dimensional matrix is effective, and this design enhances the connection of elements in the triplet; Using Shared Token instead of Span results in a decrease in the F1 value of the model, indicating that it is more effective to directly share span vectors. Ablation experiments show that all modules designed to improve the effect of relation extraction in the model are effective.

## Conclusion

In this paper, a new span-based multi-head selection model is proposed, which constructs span vector representation by span-tagger as well as span embedding, uses LSTM and multi-head self-attention for span feature extraction, uses multi-head selection mechanism to achieve single-step relation extraction, and introduces span classification to assist training, which provides span type information and span type constraints for relation extraction tasks. Compared with various baseline models, F1 scores obtained the best results on the English relation extraction dataset NYT and the Chinese relation extraction dataset DuIE2.0. Although this method can effectively solve entity overlap, error accumulation, exposure bias, etc., there is still room for exploration. It consumes a large amount of memory space when embedding spans, the high computational complexity when selecting relations, which makes it impossible to model long texts, the practice of constructing span vector representation is a bit rough, and the way to introduce span type information and span type constraints is not straightforward. Unfortunately, due to the constraints of the experimental environment, we failed to finally test how long the window length can reach the peak of the model effect. In other words, there is still room for improvement in our model. The focus of future work is how to build span vectors in

**Table 6. Comparison experiment results of mapping strategy.**

| Dataset | Ratio of $\alpha$ to $\beta$ | Prec. | Rec. | F1 |
|---|---|---|---|---|
| NYT | 1:1 | 95.7 | 80.5 | 87.5 |
| | 3:1 | 97.5 | 82.0 | 89.1 |
| | 1:3 | 97.3 | 76.6 | 85.7 |
| DuIE | 1:1 | 78.9 | 73.0 | 75.8 |
| | 3:1 | 80.9 | 73.8 | 77.2 |
| | 1:3 | 81.9 | 69.5 | 75.1 |

Other hyperparameter values: mapping = start, window_length = 6(NYT) and 15(DuIE).

**Table 7. Results of ablation experiments.**

| Dataset | Research Content | Prec. | Rec. | F1 |
|---|---|---|---|---|
| NYT | Remove Span-Classify | 92.3 | 77.1 | 84.0 |
| | Remove the Multi-head Selection | 86.1 | 75.1 | 80.2 |
| | Use Shared-Token | 95.3 | 78.6 | 86.1 |
| | SMHS | **97.5** | **82.0** | **89.1** |
| DuIE | Remove Span-Classify | 77.0 | 69.4 | 73.0 |
| | Remove the Multi-head Selection | 71.3 | 61.1 | 65.8 |
| | Use Shared-Token | 79.1 | 70.7 | 74.7 |
| | SMHS | **80.9** | **73.8** | **77.2** |

The default hyperparameter values are: mapping = start, window_length = 6(NYT) and 15(DuIE), $\alpha$:$\beta$=3:1.

a more refined and effective way, so as to optimize the time complexity and space complexity without reducing the accuracy. Besides, introducing type information and type constraints more directly and explicitly is also the direction of the subsequent research.

## Supporting information

**S1 Dataset.**
(ZIP)

## Author Contributions

**Conceptualization:** Dongchen Han, Zhaoqian Zheng.

**Data curation:** Haiting Pang.

**Investigation:** Shanshan Feng.

**Methodology:** Dongchen Han, Zhaoqian Zheng.

**Software:** Dongchen Han.

**Supervision:** Hui Zhao.

**Validation:** Zhaoqian Zheng.

**Writing – original draft:** Dongchen Han.

**Writing – review & editing:** Dongchen Han.

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
