## [Decision Letter · Decision Letter 0]

27 Sep 2022

PONE-D-22-22843Span-Based Single-Stage Joint Entity-Relation Extraction Model for Chinese Short TextsPLOS ONE

Dear Dr. Han,

Thank you for submitting your manuscript to PLOS ONE. After careful consideration, we feel that it has merit but does not fully meet PLOS ONE’s publication criteria as it currently stands. Therefore, we invite you to submit a revised version of the manuscript that addresses the points raised during the review process.

ACADEMIC EDITOR: There are two reviewers. One prefers to a major revision and another one suggests a minor revision. As a result, a major revision is suggested now. ===============================

We look forward to receiving your revised manuscript.

Kind regards,

Yiming Tang, Ph.D.

Academic Editor

PLOS ONE

4. Please amend the manuscript submission data (via Edit Submission) to include author Shanshan Feng and Haiting Pang.

Reviewers' comments:

Reviewer's Responses to Questions

**Comments to the Author**

1. Is the manuscript technically sound, and do the data support the conclusions?

Reviewer #1: Yes

Reviewer #2: Partly

2. Has the statistical analysis been performed appropriately and rigorously? 

Reviewer #1: Yes

Reviewer #2: No

3. Have the authors made all data underlying the findings in their manuscript fully available?

Reviewer #1: Yes

Reviewer #2: Yes

4. Is the manuscript presented in an intelligible fashion and written in standard English?

Reviewer #1: Yes

Reviewer #2: No

5. Review Comments to the Author

Reviewer #1: The manuscript, in its present form, contains several weaknesses. Adequate revisions to the following points should be undertaken to justify the recommendation for publication.

1. Specify the limitations and drawbacks of the proposed method.

2. The contribution is not stated also add at the end of the introduction section.

3. A deep and detailed comparison with other methods is mandatory.

4. Please add future work to the conclusion section and discuss it briefly.

Reviewer #2: The manuscript presents a neural approach for relation extraction from Chinese Short Texts. Experiments were performed on a publicly available dataset and the results were compared with other approaches from the literature. While the work seems to be promising, the following issues should be addressed before publication.

1. Is the approach specific to the Chinese language? If yes, please explain in more detail why (note that this may limit the significane of the work). If not, it would be nice to see how the presented approach works in case of other languages, i.e., experiments on other datasets would increase the scientific value of the manuscript.

2. Based on the description, it is rather difficult to reproduce the approach. Therefore, I suggest to publish the implementation (source codes) in a repository, such as github.

3. I wonder, if both hyperparameters of the loss function, alpha and beta, are needed? It seems that what actually matters is the ratio of alpha and beta, right? Currently, the Authors propose to use alpha=1.5 and beta=0.5, but I guess that we should get the exactly same results with alpha=3 and beta=1 if the learning rate is halfed. Do the Authors agree? If so, one could eliminate one of the hyperparameters out of alpha and beta for simplicity and clarity.

4. According to the first sentence of the "Dataset" subsection in the section titled "Experiments" (line 215), the Authors compare their results with previously published results. However, in the second paragraph of the same section, the Authors write that they modified both the training and the test data, so I wonder if the results on the new test set are directly comparable with the results obtained by other researchers on the original test set?

5. It seems that the Authors did not perform statistical significance tests, although they claim (at the end of the Introduction) that their approach significantly outperform other approaches from the literature.

6. The manuscript should be carefully proofread, esp. because PLOS ONE does not copyedit accepted manuscripts. Minor issues of the current version range from "unusual" (not necessarily incorrect) phrasings over inconsistent use of capital letters, to broken references. A few examples (the list is not intended to be complete!):

- " The model experimented on..."

- Currently, section titles are inconsistent in terms of the usage of capital letters: "Related Work", "MODEL", "hyperparameter comparison experiment"

- In several(!) cases, a space should be added after the dot (".") at the end of the sentence, e.g. "...sample.As shown" -> "...sample. As shown"

- Instead of "potential to be developed" and "next research", phrases like "room for improvement" and "future work" would be a more common (please remember to rephrase the entire sentence if the new phrase does not fit into the original sentence).

- Spelling of "GraphRel" and "CMAN" is not always consistent (in line 263: "Graphrel", in line 266: "cman"). I sugges to use "GraphRel" and "CMAN" throughout the entire manuscript.

- In line 260, I guess the Authors actually mean "precision" instead of "accuracy". (Note that "accuracy" usually refers to the number of correct classification decision divided by the total number of classification decisions.)

- Lines 240-242 literally repeat what is written in the previous lines.

- At the end of line 185, I can see a flipped "!" and a flipped "?" instead of "(" and ")" around t_i and t_j (it may be a rendering issue, but please double check).

- In lines 92-94: references to sections are broken.

6. PLOS authors have the option to publish the peer review history of their article (what does this mean?). If published, this will include your full peer review and any attached files.

Reviewer #1: No

Reviewer #2: **Yes: **Krisztian Buza

---

## [Author Response · Author response to Decision Letter 0]

30 Nov 2022

To Reviewer #1:

1.The limitations and shortcomings of the method are written in lines 328 “Although ...”.

2.Follow your advice and add a contribution note at the end of the introduction.

3.A detailed comparison with other methods is made in the "Validation Experiences" section.

4.Follow your suggestions and add a discussion on future work in the conclusion section.

To Reviewer #2:

1.This method is not only for Chinese. We supplement the experiment and results on NYT dataset in our paper.

2.Follow your suggestions and the code of this project has been opened to GitHub, and the address is also mentioned in the paper.

3.Following your suggestions, I changed the description of the super parameters alpha and beta in the paper. The discussion of these two parameters in the paper is based on a fixed learning rate.

4.In this paper, standard data sets are used to ensure that the comparative experiments are on the same track.

5.At the end of the introduction, some descriptions of the effects of the model are added.

6.We sincerely thank the reviewer for careful reading. As suggested by the reviewer, We have unified several cases in the paper, such as "GraphRel" and "CMAN"; Correction of punctuation errors; Modified "precision" instead of "accuracy"; Fixed rendering issues; Several wording changes were made using more common statements.

---

## [Decision Letter · Decision Letter 1]

13 Dec 2022

PONE-D-22-22843R1Span-Based Single-Stage Joint Entity-Relation Extraction ModelPLOS ONE

Dear Dr. Zhao,

Thank you for submitting your manuscript to PLOS ONE. After careful consideration, we feel that it has merit but does not fully meet PLOS ONE’s publication criteria as it currently stands. Therefore, we invite you to submit a revised version of the manuscript that addresses the points raised during the review process.

We look forward to receiving your revised manuscript.

Kind regards,

Yiming Tang, Ph.D.

Academic Editor

PLOS ONE

Reviewers' comments:

Reviewer's Responses to Questions

**Comments to the Author**

1. If the authors have adequately addressed your comments raised in a previous round of review and you feel that this manuscript is now acceptable for publication, you may indicate that here to bypass the “Comments to the Author” section, enter your conflict of interest statement in the “Confidential to Editor” section, and submit your "Accept" recommendation.

Reviewer #2: (No Response)

2. Is the manuscript technically sound, and do the data support the conclusions?

Reviewer #2: (No Response)

3. Has the statistical analysis been performed appropriately and rigorously? 

Reviewer #2: No

4. Have the authors made all data underlying the findings in their manuscript fully available?

Reviewer #2: Yes

5. Is the manuscript presented in an intelligible fashion and written in standard English?

Reviewer #2: Yes

6. Review Comments to the Author

Reviewer #2: Although the manuscript improved compared to the initial submission, e.g. I appreciate experiments on NYT dataset and the publication of the source code, I have to say that the Authors addressed my concerns only partially and their answers are exceptionally short, not always describing what exactly they changed to address the concerns.

(A) For example, I asked the Authors to perform statistical significance tests. Their answer, i.e., "At the end of the introduction, some description of the effects of the model are added" simply does not address the issue: the reason to perform statistical significance tests is to check whether the observed differences might be attributed to random fluctuations or not. As far as I can judge, statistical significance tests are essential in scientific works, or if it is not possible to perform rigorous significance testing, the reason should be explained clearly and the Authors should find other ways to justify that their conclusions drawn from the observations are not because of random fluctuations.

(B) Tab. 6 in the new version looks somewhat strange with many missing values: is it not possible to perform experiments with the configurations indicated by the "-" on the NYT dataset? Why? Is the reason conceptual or computational? Even if it is not possible, I would exclude those rows of the table that contain missing values only, such as "2:1" in case of NYT.

7. PLOS authors have the option to publish the peer review history of their article (what does this mean?). If published, this will include your full peer review and any attached files.

Reviewer #2: No

---

## [Author Response · Author response to Decision Letter 1]

23 Dec 2022

Editor-in-Chief of PLOS ONE

Dear editor and reviewers

On behalf of all the contributing authors, I would like to express our sincere appreciations of your letter and reviewers’ constructive comments concerning our article entitled “Span-Based Single-Stage Joint Entity-Relation Extraction Model” (Manuscript No.: PONE-D-22-22843R1). These comments are all valuable and helpful for improving our article. According to the associate editor and reviewers’ comments, we have made extensive modifications to our manuscript and supplemented extra data to make our results convincing. In this revised version, changes to our “Revised Manuscript with Track Changes” were all highlighted within the document by using yellow-colored text. Point-by-point responses to the nice associate editor and two nice reviewers are listed below this letter.

To Reviewer #2:

A.We are sorry that we failed to complete the statistical significance test before. In order to prove that the model results do not come from random fluctuations, we have added ablation experiments in this revision to ensure the effectiveness of each component of the model.

B.Tab 6 has been revised and redundant experiments have been removed to make it clear.

Thank you again for your positive comments on our manuscript. PLOS ONE is an influential magazine. I hope our articles can be published in your magazine. If there are any other corrections, please feel free to contact us.

---

## [Decision Letter · Decision Letter 2]

16 Jan 2023

Span-Based Single-Stage Joint Entity-Relation Extraction Model

PONE-D-22-22843R2

Dear Dr. Zhao,

We’re pleased to inform you that your manuscript has been judged scientifically suitable for publication and will be formally accepted for publication once it meets all outstanding technical requirements.

Kind regards,

Yiming Tang, Ph.D.

Academic Editor

PLOS ONE

Additional Editor Comments (optional):

Reviewers' comments:

Reviewer's Responses to Questions

**Comments to the Author**

1. If the authors have adequately addressed your comments raised in a previous round of review and you feel that this manuscript is now acceptable for publication, you may indicate that here to bypass the “Comments to the Author” section, enter your conflict of interest statement in the “Confidential to Editor” section, and submit your "Accept" recommendation.

Reviewer #2: All comments have been addressed

2. Is the manuscript technically sound, and do the data support the conclusions?

Reviewer #2: (No Response)

3. Has the statistical analysis been performed appropriately and rigorously? 

Reviewer #2: (No Response)

4. Have the authors made all data underlying the findings in their manuscript fully available?

Reviewer #2: (No Response)

5. Is the manuscript presented in an intelligible fashion and written in standard English?

Reviewer #2: (No Response)

6. Review Comments to the Author

Reviewer #2: (No Response)

7. PLOS authors have the option to publish the peer review history of their article (what does this mean?). If published, this will include your full peer review and any attached files.

Reviewer #2: No

---

## [Editor Report · Acceptance letter]

27 Jan 2023

PONE-D-22-22843R2 

Span-Based Single-Stage Joint Entity-Relation Extraction Model 

Dear Dr. Zhao:

I'm pleased to inform you that your manuscript has been deemed suitable for publication in PLOS ONE. Congratulations! Your manuscript is now with our production department. 

Kind regards, 

on behalf of

Professor Yiming Tang 

Academic Editor

PLOS ONE